# Further Molecular Diagnosis Determines Lack of Evidence for Real Seed Transmission of Tomato Leaf Curl New Delhi Virus in Cucurbits

**DOI:** 10.3390/plants12213773

**Published:** 2023-11-04

**Authors:** Cristina Sáez, Amina Kheireddine, Arcadio García, Alicia Sifres, Alejandro Moreno, María Isabel Font-San-Ambrosio, Belén Picó, Carmelo López

**Affiliations:** 1Institute for the Conservation and Breeding of Valencian Agro-Diversity, Universitat Politècnica de València (COMAV-UPV), Camino de Vera s/n, 46022 Valencia, Spain; amina08212@gmail.com (A.K.); alsifcue@upvnet.upv.es (A.S.); mpicosi@btc.upv.es (B.P.); 2Centro de Biotecnología y Genómica de Plantas UPM-INIA and E.T.S. Ingeniería Agronómica, Alimentaria y de Biosistemas, Universidad Politécnica de Madrid, 28031 Madrid, Spain; 3Instituto de Biología Molecular y Celular de Plantas, Consejo Superior de Investigaciones Científicas—Universitat Politècnica de València, Camino de Vera s/n, 46022 Valencia, Spain; arcadiogarcia@ibmcp.upv.es; 4Sakata Vegetables Europe Ltd., 04738 Almería, Spain; alejandro.moreno@sakata.eu; 5Instituto Agroforestal Mediterráneo, Universitat Politècnica de València (IAM-UPV), Camino de Vera s/n, 46022 Valencia, Spain; mafonsa@upvnet.upv.es

**Keywords:** seed transmission, begomovirus, ToLCNDV, cucurbit seedlings, qPCR, rolling-circle amplification

## Abstract

Begomoviruses (family Geminiviridae) cause serious diseases in many crop families. Since 2013, the Spanish isolate of tomato leaf curl New Delhi virus (ToLCNDV) has been a limiting factor for cucurbits production in the Mediterranean basin, forcing farmers to adapt new management and control techniques. Although it is well-known that begomoviruses are naturally transmitted by the whitefly *Bemisia tabaci*, the capacity of these viruses to be vertically transmitted through seeds remains controversial. Clarifying the potential ToLCNDV seed transmission is essential to understand the epidemiology of this threating-for-cucurbits virus and to design appropriate control strategies. We assessed ToLCNDV distribution in the leaves, flowers and seeds of the infected plants of susceptible *Cucumis melo* accessions and toleration to the infected genotypes of *Cucurbita moschata* by conventional and quantitative PCR. We analyzed whether the viral particle was transmitted to offspring. We also evaluated ToLCNDV presence in commercial seeds of cucurbits (zucchini (*Cucurbita pepo*), melon (*C*. *melo*), cucumber (*Cucumis sativus*) and watermelon (*Citrullus lanatus*)) and in their progenies. As the assayed seedlings remained symptomless, we increased the reliability and accuracy of detection in these samples by searching for replicative forms of ToLCNDV by combining Southern blot hybridization and rolling-circle amplification (RCA). However, integral genomic DNA was not identified in the plants of offspring. Although the seedborne nature of ToLCNDV was confirmed, our results do not support the transmission of this virus from contaminated seeds to progeny.

## 1. Introduction

Cultivated Cucurbitaceae family species include vegetables and fruits that supply the current diets of diverse cultures worldwide with essential vitamins and minerals. Spain is one of the leading world producers and the first European exporting country of cucurbits [1]. Traditionally, the production of these crops has been severely affected by viral cucurbit diseases, with infections of aphid-borne viruses belonging to the Potyviridae family as the most widespread and damaging [2,3]. More recently, whitefly-transmitted viruses (family Geminiviridae) have been discovered in different cucurbit species [4,5]. One of these viruses, the tomato leaf curl New Delhi virus (ToLCNDV), a member of the genus Begomovirus of the family Geminiviridae, was detected in Spain in 2012. It rapidly spread in the southern regions of this and neighboring countries in the Mediterranean basin. The subsequent epidemics had a significantly negative impact on agriculture with substantial losses in this horticultural region [6].

The ToLCNDV genome consists of two circular single-stranded DNA molecules of approximately 2.7 kb each (designated as DNA-A and DNA-B), both required for essential viral functions and encapsidated in geminate particles [7,8]. ToLCNDV was first detected on tomato (*Solanum lycopersicum* L.) in north India in 1995 [9]. Later, it propagated to other Asian countries on several hosts, particularly vegetable species of the Solanaceae and Cucurbitaceae families [10,11]. The populations of the isolate that emerged in the Mediterranean basin, designated as ToLCNDV-ES, present a very low degree of genetic variability [12,13] and affect mainly cucurbits, preferentially zucchini (*Cucurbita pepo* L. subsp. *Pepo*), melon (*Cucumis melo* L.) and cucumber (*Cucumis sativus* L.) [14,15,16,17,18]. Symptoms include curling and the severe mosaic of young leaves, shorter internodes and smaller-sized fruit with skin deformations [14], which often result in null or reduced yields and a lower market value.

Understanding the epidemiology and evolution of plant virus diseases constitutes a fundamental strategy to contrlolling devastating viruses like ToLCNDV for cucurbit crops. One key aspect for the success of disease outbreaks is the efficiency and mode of transmission of the causal agent. For plant viruses, two main modes of transmission have been described: horizontal, in which the virus is transmitted from plant to plant between individuals, which can be from the same generation, and vertical, in which viral transmission occurs from parent to offspring, mostly through seeds [19]. Like most begomoviruses, ToLCNDV is horizontally transmitted only by the whitefly *Bemisia tabaci* (Gennadius) in a circulative persistent manner [7,20,21,22]. Experimentally, the cloned genomic DNAs of this virus can be used as infectious constructions in *Agrobacterium*-mediated transmission (agroinoculation). In addition, some isolates of this virus, including ToLCNDV-ES, can be mechanically sap-transmitted to a narrow host range [20,23,24,25]. However, these modes of transmission have not been demonstrated under natural conditions. This leaves vertical transmission as the only alternative for the whitefly-mediated transmission of ToLCNDV-ES. 

Until recently, the seed transmission of geminiviruses had been considered inefficient, but accumulated evidence challenges this traditional view [26]. For instance, the sweet potato leaf curl virus (SPLCV, genus Begomovirus) has been detected in seeds from SPLCV-infected sweet potato (*Ipomoea batatas* (L.) Lam.) plants, and the transmission rate of SPLCV from seeds to seedlings went up to 15% [27]. The mung bean yellow mosaic virus (MYMV, genus Begomovirus) has been encountered in seeds of black gram (*Vigna mungo* L. Hepper) plants naturally infected in the field, and the virus was also detected in 32% of the seedlings of offspring [28]. The beet curly top virus (BCTV) and the beet curly top Iran virus (BCTIV) (both of genus Curtovirus) have been detected in 38.2–78.0% and 8.8–18.5% of the seedlings that developed from the seeds of a petunia cultivar (*Petunia x hybrida* hort. ex E. Vilm.) infected with BCTV and BCTIV, respectively [29]. Tomato yellow leaf curl virus (TYLCV) has also been found in seeds from TYLCV-infected tomato plants, and the reported average vertical transmission rate was 80% [30], as well as in white soybean (*Glycine max* (L.) Merr.) and sweet pepper (*Capsicum annuum* L.) [31,32]. Indian isolates of ToLCNDV were reported as being seed-transmitted for Indian isolates in chayote (*Sechium edule* L.) in Tamil Nadu, India [33], in cucumber in Taiwan [34], in bitter gourd (*Momordica charantia* L.) in Coimbatore, India [35] and in sponge gourd (*Luffa cylindrica* L.) in Varanasi, India [36]. In cucurbits, the seed transmission of the Spanish ToLCNDV isolate has also been investigated in Mediterranean regions. Kil et al. [37] reported vertical ToLCNDV-ES transmission through seeds in zucchini plants in Italy, with transmission rates exceeding 60% in the evaluated offspring seedlings. In contrast, other works have observed no evidence for TYLCV seed transmissibility in *Nicotiana benthamiana* [38]. Likewise, recent studies on tomato have reported the absence of TYLCV and tomato yellow leaf curl Sardinia virus (TYLCSV) transmission through infected seeds in tests performed at different locations [39,40]. Finally, the seed transmission of SPLCV was not detected after the recent large-scale screening of sweet potato plants [33,34,35,36,41]. The analysis of melon seedlings germinated from the ToLCNDV-ES-infected seeds did not support vertical transmission [42]. So, whether geminiviruses are seed-transmitted is still a matter of debate.

Elucidating whether geminiviruses like ToLCNDV-ES are seed-transmitted is essential to define import requirements and to develop and implement control strategies. For instance, the EFSA Panel on Plant Health [43] considered that even though ToLCNDV transmission through seeds may be possible, the production of commercial seeds contaminated by ToLCNDV is not very likely. Consequently, seed health tests for this virus have not yet been implemented, a situation that would change if efficient seed transmission was demonstrated. Hence, our research goal was to evaluate whether ToLCNDV-ES can be transmitted through zucchini, melon, pumpkin, cucumber and watermelon (*Citrullus lanatus* (Thunb.) Matsumara and Nakai) seeds by assaying house and commercially produced seeds, which could serve as a primary source of inoculum for transmission by vectors in the field. Moreover, as the use of resistant host varieties in cucurbits is the most efficient viral disease management strategy for ToLCNDV-ES-induced diseases [44], our trials included the accession Nigerian Local (*Cucurbita moschata* (Duchesne), originally from Nigeria), which is tolerant to ToLCNDV-ES [45]. In this way, the impact of tolerance to virus infection on seed transmission could be analyzed. The results herein obtained contribute to identifying potential ToLCNDV-ES primary sources of inoculum, with an impact on both global seed trade and germplasm conservation in gene banks. 

## 2. Materials and Methods

### 2.1. Obtaining Seeds from ToLCNDV-Infected Cucurbit Plants and Sampling

#### 2.1.1. Seeds Obtained in the COMAV-UPV Greenhouse

In order to investigate the seed transmission of ToLCNDV-ES in the Cucurbitaceae family, we tested the offspring of 27 genotypes belonging to the *C. melo* and *C. moschata* species. The number of assayed genotypes and their botanical classification are shown in Table 1.

Seeds were provided by the Institute for the Conservation and Breeding of Agricultural Biodiversity genebank (COMAV-Universitat Politècnica de València, Valencia, Spain). All the *C. melo* genotypes had been previously reported as susceptible to ToLCNDV [23]. The three evaluated *C. moschata* plants were the pumpkin accession Nigerian Local, known to be resistant to ToLCNDV when the virus is mechanically transmitted and to be tolerant when the virus is vector-inoculated by whiteflies [45], and two breeding lines with different resistance levels that derive from the Nigerian Local accession. Melon seed coats were slightly opened by forceps to facilitate germination. All the seeds of each genotype were disinfected by soaking them in 10% solution of sodium hypochlorite for 3 min and washing them for 5 min in distilled water. Germination was performed on Petri plates with moistened cotton at 37 °C for 48 h. Seedlings were transplanted to pots in a growth chamber under controlled environmental conditions of 25 °C, 60% relative humidity and a 16–8 h light/dark photoperiod. In the four true-leaf growth stages, seedlings were transplanted to a leak-proof greenhouse.

For inoculation purpose, a ToLCNDV-ES-infectious clone was agro-infiltrated by injection into petioles of MU-CU-16 zucchini (*C. pepo*) plants, as described in Sáez et al. [45]. After noting visible ToLCNDV infection symptoms, these plants and a population of whiteflies were established in the same aforementioned greenhouse to constitute the source of inoculum and the natural vector to perform ToLCNDV transmission to the selected genotypes. Upon inoculation, healthy melon and pumpkin plants were in the flowering developmental stage. These plants were later monitored for ToLCNDV infection by symptom development, and virus presence was confirmed by PCR, as described in López et al. [23]. Differences in viral distribution through plants were further investigated by qPCR in the leaf and flower tissues of the selected genotypes, as described below. All the infected plants were self-pollinated, and fruits were cultivated and harvested at maturity. Seeds were collected and dried to be preserved at 4 °C. To explore virus location, three seeds from the infected fruits of the selected melon genotypes were used to separately test the internal (embryo and endosperm) and external (testa/coat) seed parts for viral infection. In each seed part, viral titers of seeds were determined by qPCR and comparing between the untreated and surface-disinfected seeds with sodium hypochlorite, as described above. Finally, 15 non-surface-disinfected seeds per fruit were germinated and grown in a growth chamber in three batches of five plants each to evaluate ToLCNDV seed transmission to the progeny.

#### 2.1.2. Analysis of the Seeds from a Commercial Greenhouse

Both environmental factors and host genotype influence the seed transmission of viruses [46,47]. To consider these effects, ToLCNDV seed transmission was additionally evaluated using 60 seeds from the ToLCNDV-diseased melon fruits cultivated in a commercial greenhouse located in La Mojonera (Almería, Spain). These fruits came from different varieties from those used in the COMAV greenhouse experiments and were grown under different conditions. Four of these seeds were reserved to perform a PCR test of whole seeds or by separating embryo–endosperm and coat to evaluate ToLCNDV accumulation. The remaining 56 seeds were germinated as described above but avoiding previous disinfection. Seedlings were transplanted to pots and cultivated in a climatic chamber under the same conditions. 

#### 2.1.3. Analysis of the Seeds from a Commercial Nursery

Finally, a third assay was performed with the commercial seeds to evaluate if it was possible to detect ToLCNDV in them and, in the positive cases, to test seed transmission. The batches with 15 seeds from the 43, 17, 30 and 19 varieties of cucumber (*C. sativus* var. *sativus*), melon (*C. melo* subsp. *melo*), watermelon (*C. lanatus* var. *lanatus*) and zucchini (*C. pepo*), respectively, were bought from a commercial nursery. Five whole seeds of each variety were checked by PCR to identify ToLCNDV contamination (as described below). In the event of ToLCNDV detection, the remaining seeds of the corresponding batch were used to evaluate vertical ToLCNDV transmission to offspring.

### 2.2. Evaluation of ToLCNDV Transmission to Offspring through Seeds

By avoiding previous disinfection, the seeds obtained from the infected plants were germinated and seedlings were grown as described above. All the seedlings were weekly monitored to detect symptoms development. At 30 and 60 days post germination (dpg), the completely expanded youngest leaf of each seedling from the assays of Section 2.1.1 was sampled and used for DNA extraction, followed by ToLCNDV detection by conventional PCR and qPCR testing. 

To assess additional temporal development stages, the seedlings of the offspring from both the commercial greenhouse plants and the seeds bought in a nursery were weekly sampled from 2–4 weeks after germination (wag). In both studies, the tissue from the apical leaf of each plant was collected and subsequently used for DNA extraction and qPCR.

### 2.3. DNA Extractions from the Different Plant Tissues, Seeds and Seedlings to Make a ToLCNDV Diagnosis

The total genomic DNA for the ToLCNDV diagnosis was extracted from the parental leaf and flower (petals, stamens, pistils) tissues and from the leaves of seedlings by the Cetyltrimethyl ammonium bromide (CTAB) method [48]. To identify ToLCNDV infection in flower tissue, bulks of two or three flowers of the same plant were used for DNA extraction. DNA was quantified using a NanoDrop 1000 spectrophotometer and diluted with sterile distilled deionized water to a final concentration of 50 ng/μL. 

To avoid coprecipitated polysaccharides and inhibitors of the PCR reaction, the DNA of whole seeds, coats and internal seed tissues (embryos and endosperms) was extracted using Zymo-Spin™ I columns (Zymo Research, Irvine, CA, USA). Each seed or seed component was mashed in an Eppendorf tube with stainless steel “UFO” beads of 3.5 mm (Next Advance, Inc., Raymertown, NY, USA), which have sharper edges that are specific for resilient samples, in a Retsch (MM300) homogenizer for 1.5 min at 30 s^−1^. The mashed samples were centrifuged and placed on ice. To each tube 1.4 mL of extraction buffer (4 M guanidine thiocyanate; 0.1 M sodium acetate, pH 5.5; 10 mM ethylenediaminetetraacetic acid (EDTA), 0.1 M 2-mercaptoethanol) was added and mixed vigorously in a vortex. After 5 min of centrifuging at 13,000× *g* rpm, 700 μL of the supernatant was transferred to silica gel spin columns and then washed twice with 500 μL of washing buffer (70% ethanol, 10 mM sodium acetate, pH 5.5). Finally, the total DNA was eluted by loading 10 μL of 20 mM Tris-HCl, pH 8.5, which was stored at −20 °C.

### 2.4. Detection of ToLCNDV by PCR

The presence of ToLCNDV in the plant samples was determined by a PCR reaction to detect the presence of both viral components using two specific pairs of primers. For this purpose, 1 μL aliquots of total DNA (50 ng) were used as templates in the PCR reactions of 20 μL with 1 U of DreamTaq DNA polymerase (Thermo Scientific™, Waltham, MA, USA), 1 µM of two different primer pairs (To-A1F/To-A1R, and To-B1F/To-B1R), 10× DreamTaq Buffer (Thermo Scientific™) and 0.2 mM dNTPs and water up to the final volume. The two primer pairs were derived from Spanish isolate Murcia 11.1, one from segment DNA-A, accession number KF749225 (To-A1F 5′-GGGTTGTGAAGGCCCTTGTAAGGTGC-3′, position 476–501 and To-A1R 5′-AGTACAGGCCATATACAACATTAATGC-3′, position 954–979) and another one from segment DNA-B, accession number KF749228 (To-B1F 5′-GAAACACAAGAGGGCTCGGA-3′, position 637–656, and To-B1R 5′-GCTCCACTATCAAAGGGCGT-3′, position 1294–1313). The cycling conditions consisted of incubation at 94 °C for 5 min and 35 cycles of 95 °C for 30 s, 55 °C for 30 s and 72 °C for 45 s, with a final extension of 10 min at 72 °C. The resulting PCR products of 504 and 677 bp in length were analyzed by electrophoresis in 1.5% agarose gels in TAE buffer (40 mM Tris, 20 mM sodium acetate and 1 mM EDTA, pH 7.2) and stained with ethidium bromide. To verify the identity of the amplified fragments, the PCR products of the random samples were sequenced in both directions by an ABI PRISM DNA Sequencer 377 (Perkin–Elmer, Waltham, MA, USA) using the same primers as for the amplification reaction. The presence of intact DNA in the negative seed samples was confirmed by PCR to amplify the β-actin gene in the cucumber, melon and watermelon (CIACT-F 5′-CCATGTATGTTGCCATCCAG-3′, CIACT-R 5′-GGATAGCATGGGGTAGAGCA-3′) samples and CpACS7 in the Cucurbita seeds (CpACS7F 5′-GTGAGAGTGGCAAGAGGGAG-3′, CpACS7R 5′-CGGCATTGCAAAGAAAAAGCAG-3′).

### 2.5. Relative Viral Titer Estimation by Quantitative PCR (qPCR)

To estimate the ToLCNDV titers in the different analyzed samples, the relative quantification of viral loads by qPCR was also performed at 60 days after planting (dap) in the selected positive leaf and flower samples from the whitefly-infected plants at COMAV-UPV. The seeds and seedling samples that tested positive for ToLCNDV presence were also subjected to qPCR relative quantification. For this purpose, total DNA was diluted to a final concentration of 5 ng·μL^−1^. Equal amounts of genomic DNA (15 ng) were used as templates in the qPCR reactions of 15 μL, containing 7.5 μL of 2× iTaqTM universal SYBR^®^ Green Supermix, 1.5 μL of each primer and 1.5 μL of H_2_O. Primers ToLCNDVF1 (5′-AATGCCGACTACACCAAGCAT-3′, positions 1145–1169) and ToLCNDVR1 (5′-GGATCGAGCAGAGAGTGGCG-3′, positions 1399–1418) were employed for the amplification of a 274 bp fragment of viral DNA-A. As reference host genes, single-copy genes CmWIP1 (primers CmWIP1F (5′-TAGGGCTTCCAACTCCTTCCTCTT-3′) and CmWIP1R (5′CTTGCAATTGATGGGTGTGATCTTCTTG-3′) and CpACS7 were amplified in the melon and *C. moschata* samples, respectively. Quantification was performed in a LightCycler^®^ 480 Instrument (Roche, Basel, Switzerland) under cycling conditions that consisted of an incubation step at 95 °C for 15 min, followed by 45 cycles of 95 °C for 5 s, 57 °C for 30 s and 72 °C for 30 s. Finally, a melting curve was obtained for each sample to confirm and characterize amplicons. Relative ToLCNDV accumulation was calculated for each sample by applying the simplified expression (2^−ΔCt^) of the method developed by [49]. Significant differences between samples were evaluated by ANOVA multiple range tests (least significance difference; LSD) using the STATGRAPHICS ^®^ Centurion XVIII (Statgraphics Technologies, Inc., The Plains, VA, USA) statistical software.

### 2.6. Southern Blot Hybridization

In order to understand how the virus is transmitted through seeds, we attempted to detect replicative forms of ToLCNDV by Southern blot in the seedling samples in which viral load was detected. As controls, the positive leaf and flower samples of the parental ToLCNDV-infected plants were included. An amount of 15 µg of total DNA of each sample was loaded onto 1% agarose gel. Electrophoresis was performed in 1× TAE buffer at 50 V for 3 h and then blotted onto a positively charged nylon membrane (Roche, Basel, Switzerland). Membranes were fixed by UV irradiation (700 × 100 mJ/cm^2^) and hybridized with a digoxigenin-labeled RNA probe, as described in Sáez et al. [50]. The results were visualized in the films exposed to membranes for 1 h at room temperature.

### 2.7. Rolling-Circle Amplification (RCA)

The total DNA of the samples of the offspring seedlings, which were positive for ToLCNDV infection, was diluted to 10 ng/µL and used as a template in the RCA by resuspending 1 µL in 10 μL of sample buffer from the TempliPhi kit (Cytiva 25-6400-10, Sigma Aldrich, St. Louis, MO, USA) and incubating at 95 °C for 90 s. After freezing on ice, a mix of 5 μL of reaction buffer and 0.2 μL of the enzyme was added to each sample. RCA reactions were carried out at 30 °C for 18 h. Subsequently, the enzyme was heat-inactivated at 65 °C for 10 min. Products were directly subjected to PCR amplification and electrophoresis separation on agarose gel, as described above. The samples of the ToLCNDV-infected leaves and reproductive organ tissues were added as positive controls, while a healthy seedling sample was included as a negative control. 

## 3. Results

### 3.1. Symptoms Development and ToLCNDV Detection in Vegetative and Reproductive Tissues of Infected Plants

*C. pepo* plants were used as a source of inoculum for ToLCNDV whitefly transmission to *C. melo* and *C. moschata*. All the *C. pepo* plants agro-infiltrated with the ToLCNDV clones developed typical curly and mosaic symptoms in young leaves after 10–15 days post inoculation (dpi). As a consequence of high disease severity, zucchini plants underwent severe growth restriction and failed to produce any fruit. All the plants died before fructification and were not included in seed transmission tests. Three weeks following the establishment of whiteflies, all the melon plants’ youngest leaves showed typical curling, yellowing and severe mosaic symptoms of ToLCNDV infection (Figure 1A). In late disease stages, both male and female melon flowers presented petal discoloration and deformation, and floral ovaries were cracked in some plants (Figure 1B,C). Despite the presence of cracked fruit with nonviable seeds (Figure 1D), 17 of the 24 plants produced melons with skin roughness symptoms but containing viable seeds (15 from the inodorus group, one from the cantalupensis group and one from the flexuosus group) (Table 1) (Figure 1E,G).

Of the *C. moschata* genotypes, the plants from accession Nigerian Local remained symptomless, and the two derived lines showed slight symptoms on leaves after each plant generated one fruit with well-formed seeds (Table 1) (Figure 1H–K). 

ToLCNDV was detected by polymerase chain reaction (PCR) in all the leaf tissues of the assayed plants (Figure 2, Table 2). As expected, the viral load detected in *C. melo* was higher than in the *C. moschata* genotypes. In almost all the male and female melon flowers, ToLCNDV-specific bands were detected on petals, stamens and pistils (Table 2). In all the genotypes in which DNA had been successfully extracted from seed coats, ToLCNDV presence was confirmed in this seed part, while the virus was not detected in the endosperms of some *C. melo* seeds.

Even in the symptomless *C. moschata* plants, ToLCNDV was detected in all the floral tissues (Table 2). In this host, ToLCNDV presence was confirmed in all the tested seed coats but in none of the endosperms (Table 2). The sequencing of the PCR products confirmed viral infection.

### 3.2. ToLCNDV Detection by Quantitative PCR in Different Parts of Inoculated Plants, Seeds and Seedlings

To quantify the differences in ToLCNDV distribution between the analyzed tissues in both the *C. melo* and *C. moschata* plants, a quantitative PCR (qPCR) assay was performed. ANOVA and LSD tests did not identify any significant differences between the means of the viral accumulation in the *C. melo* leaves and floral tissues (*p* ≤ 0.05) (Figure 3A). On average, the relative viral load in the leaves, petals and reproductive tissues of *C. moschata* was six orders of magnitude lower than that quantified in the same tissues of *C. melo* (Figure 3A). In the melon seeds obtained by self-pollination, viral titers were detected in all the seed parts (Figure 3B). However, ToLCNDV accumulation significantly reduced in the bleach-treated endosperm and embryo tissues compared with the untreated seeds or the treated seed coats (Figure 3B).

### 3.3. Evaluation of ToLCNDV Presence in the Seedlings Obtained from the Inoculated Plants

None of the offspring seedlings derived from the fruit of the ToLCNDV-infected plants at COMAV-UPV developed symptoms during the assay. Tissue samples were collected from each seedling at 30 and 60 dpg to assess ToLCNDV presence by PCR. Of the 255 evaluated *C. melo* plants, specific bands were detected in only 6 and 13 plants from the inodorus group at 30 and 60 dpg, respectively. In none of the six plants where viral load was detected at 30 dpg was the presence of the virus confirmed at 60 dpg. ToLCNDV was not detected in any plant from the *C. melo* cantalupensis and flexuosus groups. Of the 45 evaluated *C. moschata* seedlings, 7 and 19 tested positive for ToLCNDV at 30 and 60 dpg, respectively. Only in four seedlings was ToLCNDV detected at both 30 and 60 dpg.

To further assess virus transmission from seeds to offspring, the viral load in all the *C. melo* and *C. moschata* seedlings that tested positive to ToLCNDV was quantified by conducting a qPCR analysis. On average, relative ToLCNDV accumulation was lower in the *C. melo* plants (means of 2^−ΔCt^ = 29.19 ± 13.66 and 2^−ΔCt^ = 17.88 ± 6.0 at 30 and 60 dpg, respectively) than in the *C. moschata* plants (means of 2^−ΔCt^ = 191.64 ± 68.92 and 2^−ΔCt^ = 96.14 ± 40.11 at 30 and 60 dpg, respectively). The *C. moschata* seedlings harbored similar viral loads as the parental leaf tissues (*p* ≤ 0.05), while the viral titers detected in the *C. melo* plant progeny were, on average, 10^5^ times lower than for the mother plant leaves (*p* ≤ 0.05). 

### 3.4. Evaluation of ToLCNDV Presence in the Seeds from a Commercial Greenhouse and the Seedlings of Offspring

ToLCNDV was detected in three of the four evaluated seeds when whole seeds or separated coats were analyzed. Endosperm and embryos were negative. In 6 seedlings of the 56 germinated to assess vertical ToLCNDV transmission, the virus was detected by both PCR and qPCR 4 wag, but in only 1 of them was viral load also detected 1 week later. In three additional seedlings, ToLCNDV was detected at 5 wag.

### 3.5. Evaluation of ToLCNDV Presence in the Commercial Seeds from a Nursery and the Seedlings of Their Progeny

When the commercial seeds bought from a nursery were analyzed, ToLCNDV was not detected in the watermelon seeds (Table 3). However, in one cucumber and in one zucchini seed, ToLCNDV could be amplified by PCR (Table 3). All the analyzed seeds of one of the Piel de Sapo melon varieties tested positive for viral infection. After germinating the remaining seeds of each variety in which ToLCNDV was confirmed, all the plants remained symptomless until 5 wag. Viral presence was detected in only one melon plant (qPCR cycle threshold = 30.95).

### 3.6. Replicative Forms of ToLCNDV Were Not Identified in Any Seedling of Offspring

The samples of the seedlings in which ToLCNDV was detected, which came from the fruit obtained at COMAV-UPV or originated for commercial purposes, were subjected to Southern blot hybridization. Although a specific RNA-probe complementary to the ToLCNDV coat protein gene was used, none of the genomic forms corresponding to viral DNA were detected in the seedlings where the virus was previously identified by conventional PCR or qPCR (Figure 4A). Conversely, in the leaf, flowers and seed samples from the tissues from the infected mother plants, characteristic open circular, supercoiled and single-strand viral replicative forms were identified (Figure 4A). 

To avoid sensitivity limitations of the detection method, we included a second assay to simply detect the single- or double-stranded circular DNA of ToLCNDV. Thus, circular DNA was exponentially amplified as a previous step by a rolling-circle amplification (RCA) reaction (Figure 4B). The resulting products were used as templates in the conventional PCR assay to detect both the A and B genomic particles. The RCA products tested positive for ToLCNDV in the leaf samples, all the sampled flower parts and the different seed parts (Figure 4C). Even in the disinfected endosperm and embryo complex of the melon seeds, both the DNA-A and DNA-B of ToLCNDV were detected. Instead, no unbroken genomic components of the virus were identified in the leaves of the seedlings that germinated from the infected seeds (Figure 4C).

## 4. Discussion

Plant viruses are intracellular parasites that may be introduced into plant cells when insects transmit them during the plant-feeding process either by contact or through seeds. Around 80% of all known plant viruses are vector-borne and horizontally transmitted [51], while at least 25% of the viruses that infect plants are vertically transmitted through seeds to the seedlings of offspring [52]. Within seeds, viruses remain for lengthy periods of time and travel long distances by overcoming adverse conditions and initiating emerging epidemics, which entail serious consequences for global trade and international germplasm exchanges [53]. Consequently, detecting viral presence in seeds and determining vertical transmission through gametes and embryos to seedlings of subsequent generations are increasingly gaining importance [54,55,56].

In the diseases generated by geminiviruses, the transmission of viral particles directly into the plant phloem through whiteflies has been described as the main epidemiological factor that contributes to disease propagation. However, research into seeds as another source of geminivirus inoculum has reported some evidence for the possible seed-borne nature of certain begomoviruses in different host plants [57,58]. Yet, whether geminiviruses are seed-transmitted is still a matter of debate [39]. During real seed transmission, plant viruses infect the embryo of fertilized seeds and later the seedlings of the next generation, whereas seed-borne viruses may invade any seed tissue. Both cases require viruses to reach the reproductive organs of infected plants, and many studies have reported high geminiviral loads in plant flowers and fruits [59], as well as the presence of these viruses in whole seeds [35]. Accordingly in this work, ToLCNDV was detected in different parts of male and female flowers. Moreover, despite sodium hypochlorite disinfection, virus contamination was identified in whole seeds, coats and the ensemble of endosperms and embryos. However, the observed reduction in ToLCNDV accumulation between leaves and pistils/seeds evidences how plants rely on diverse barriers to avoid viral achievement of floral and seminal tissues. For instance, the cell death of the suspensor structure that connects the embryo and maternal tissues is programmed once the embryo has properly formed [60], which leaves viruses a narrow time window to invade this seed part. Reduced and sporadic cucumber green mottle mosaic virus (CGMMV) presence has been reported in the ovaries and ovules of cucumber plants compared with other flower tissues, which has also been associated with internal barriers that limit viral propagation [61]. The efficiency of these barriers in impairing the viral invasion of gametes and embryos is environmental-, cultivar- and viral-isolate-dependent [52]. 

With ToLCNDV, most reports that describe transmission through seeds have focused on the Asiatic strains of the virus [33,34,35,36]. Before the present work, only two research works had studied the seed transmission of ToLCNDV-ES in cucurbits, of which only one described that this mode of transmission is possible in zucchini plants [37]. The Spanish ToLCNDV isolate is adapted to infect cucurbits better than solanaceous crops [62], with *C. pepo* species being the most susceptible hosts to this viral disease [45]. The high susceptibility of zucchini plants to ToLCNDV-ES might be determined by host genetic determinants, which are also likely involved in or promote viral seed transmission. As we were unable to obtain viable seeds from the infected *C. pepo* plants, this issue has not yet been addressed. However, our results support lack of evidence for ToLCNDV-ES seed transmission in melon plants and agree with Fortes et al. [62] about the seed-borne nature of this virus in cucurbits.

Some other ToLCNDV-ES modes of transmission have been described as cultivar-dependent [63]. For instance, the whitefly transmission of the Spanish strains from zucchini to watermelon and wild cucurbit species has been reported as inefficient, even at high inoculum pressure [12,64]. Additionally, the Spanish isolate of this viral species can be mechanically transmitted to cucurbits but at different rates, depending on the assayed host species [23]. In many pathosystems, the horizontal transmission of parasites increments virulence, while vertical transmission promotes reduced virulence to allow higher host progeny production and to, therefore, optimize pathogens because they are, thus, transmitted to a higher proportion of offspring given that the host suffers less damage [65]. Then, the strictly vertically transmitted parasites tend to evolve toward slight or no virulence [66]. In the geminiviridae family, one isolate of yellow mosaic virus (YMV) vertically transmitted to greengram (*Vigna radiata* (L.) R. Wilczek) produced latent symptomless infections, with unsuccessful horizontal transmission by sap inoculation [67]. Based on this hypothesis, in the only seed transmission report about ToLCNV-ES in zucchini crops [37], all the seedlings of offspring were asymptomatic, even though the *C. pepo* species plants represent the hosts with the highest susceptible response to this isolate. Despite the zucchini seeds used by [37] being harvested from the fruit of open-field-cultivated plants displaying viral compatible symptoms, the only confirmation for viral infection to ToLCNDV was performed by PCR, but most of the viral genomic sequence remains unknown. As the mechanical transmission of a geminivirus can be determined by a single amino acid substitution in the viral movement protein [68], the vertical transmission of viral isolates might also be influenced by a few nucleotidic changes in genomes. Therefore, whether ToLCNDV transmission in zucchini plants is associated with viral or host genetic determinants must be further investigated.

Although the sudden spread of ToLCNDV in cucurbit crops in the Mediterranean basin is hardly explainable, this is not the first time that an Asian endemic virus spreads to other distal niches in the world [69,70]. Although both *B. tabaci* and contaminated germplasm propagations are the most plausible mode to mediate ToLCNDV-ES outbreaks, the results obtained in this work do not support the assumption that transmission to seedlings of subsequent generations through seeds is a likely event for this virus. Notwithstanding, the information herein provided increases epidemiological knowledge about ToLCNDV-ES disease in cucurbits, which may be considered with a view to establishing phytosanitary policies that guarantee virus-free plant material.

## 5. Conclusions

Clarifying the potential for seed transmission of ToLCNDV is essential to understand the epidemiology of this threating-for-cucurbits virus and to design appropriate control strategies. In the present study, the seed-borne nature of ToLCNDV-ES was observed in the seeds obtained from cucurbit-infected plants. However, integral genomic DNA was not identified in plants of the offspring; so, our results do not support transmission of this virus from contaminated seeds to their progeny.

## Figures and Tables

**Figure 1 plants-12-03773-f001:**
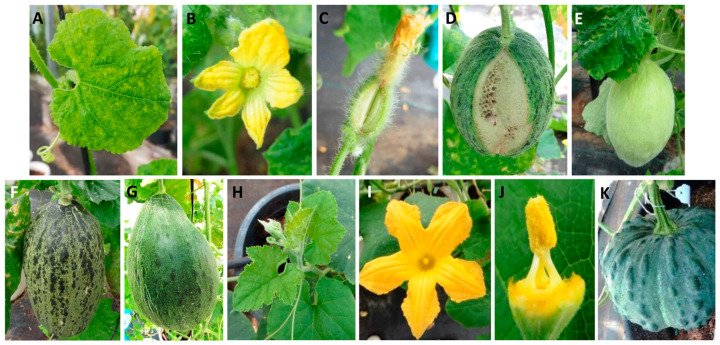
Symptom development of ToLCNDV in vegetative and reproductive organs of *C. melo* and *C. moschata* plants: (**A**) melon leaf infected with ToLCNDV displaying severe mosaic and mild curling. (**B**,**C**) male and female flowers of *C. melo*-infected plants with deformed petals and cracked ovary. (**D**) immature melon fruit aborted as consequence of ToLCNDV infection. (**E**–**G**) melon fruits with skin roughness in ToLCNDV-infected plants that arrived at a ripening stage and produced viable seeds. (**H**–**K**) leaves, flowers and fruit of asymptomatic *C. moschata* plants.

**Figure 2 plants-12-03773-f002:**
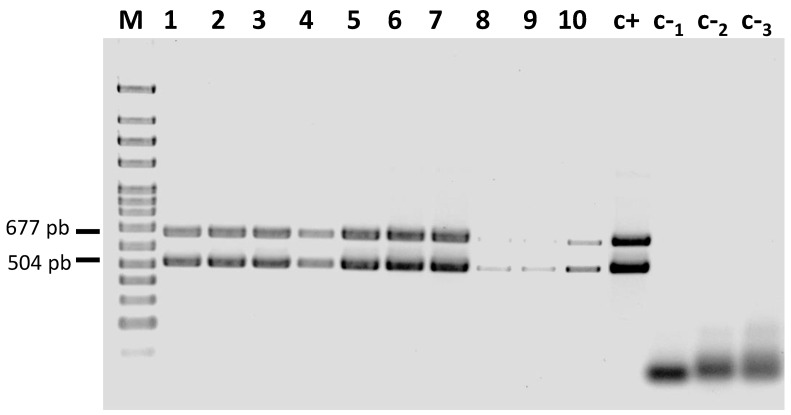
Detection of ToLCNDV-A- and -B-specific fragments by conventional PCR in symptomatic leaves of plants infected by whitefly transmission 21 days after vector and source of inoculum (MU-CU-16 zucchini plants) establishment. Lanes 1 to 5: *C. melo* inodorus group samples, lane 6: *C. melo* cantalupensis group sample, lane 7: *C. melo* flexuous group sample, lane 8: *C. moschata* Nigerian Local genotype, lanes 9 and 10: Nigerian Local derived breeding lines, c+: positive control, c-_1_ and c-_2_: healthy plants used as negative controls, c-_3_: water negative control, M: molecular weight marker DNA Ladder (NZYDNA Ladder VII, ranging from 100 to 3000 bp, NZYTECH, Lisbon, Portugal).

**Figure 3 plants-12-03773-f003:**
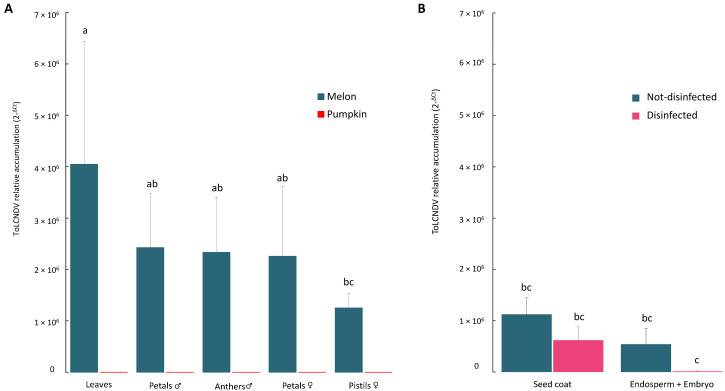
ToLCNDV relative accumulation (2^−ΔCt^) quantified by qPCR on leaf tissue, petals and reproductive organs of male (♂) and female (♀) flowers of infected plants at the greenhouse of COMAV-UPV (**A**) and in different parts of seeds obtained by their self-pollination with or without bleach treatment (**B**). Error bars represent the standard errors of the means. Bars with the same lowercase letters show not significantly different means at *p* ≤ 0.05.

**Figure 4 plants-12-03773-f004:**
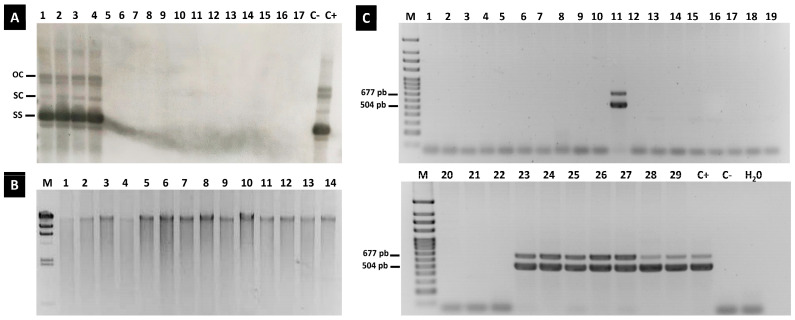
Analysis of ToLCNDV replication in seedlings of the progeny: (**A**) Southern blot analysis. Detection of open circular dsDNA (OC), supercoiled dsDNA (SC) and ssDNA (SS) forms of ToLCNDV in leaves (lane 1), flower tissues (lanes 2 and 3) and whole seeds (lane 4) of melon plants cultivated at COMAV-UPV; none of the forms were detected in seedlings of the offspring (lanes 5 to 17); C−: DNA of a healthy melon plant was used as a negative control; C+: DNA of a MU-CU-16 agro-infiltrated plant was used as a positive control. (**B**) RCA products obtained from DNA extracts of positive-for-ToLCNDV seedlings of the different offsprings evaluated (lanes 1 to 7) and tissues of parental plants inoculated with the virus (lanes 8 to 14). M: molecular weight marker DNA (Lambda DNA/*Hind*III Marker, ranging from 564 to 23,130 pb, ThermoFisher Scientific, Waltham, MA, USA). (**C**) PCR products amplified from DNA samples subjected to RCA of positive-for-ToLCNDV seedlings of the different melon offsprings evaluated (lanes 1 to 10 and 12 to 22), seed endosperm + embryo disinfected with bleach (lane 11), whole melon seed (lane 23), reproductive organs of melon flowers (lanes 24 to 27) and leaves (lanes 28 and 29) infected with ToLCNDV; C+: DNA (not subjected to RCA) of a MU-CU-16 agro-infiltrated plant was used as a positive control; C−: DNA (not subjected to RCA) of a healthy melon plant was used as a negative control; H_2_O: water was used as negative control for PCR. M: molecular weight marker DNA Ladder (NZYDNA Ladder VII, ranging from 100 to 3000 bp, NZYTECH, Lisbon, Portugal).

**Table 1 plants-12-03773-t001:** Accessions of Cucurbitaceae family infected with ToLCNDV and used to obtain progeny.

Genus	Species	Subspecies	Group	Number of Evaluated Genotypes	Number of Genotypes with Fruit
Cucumis	*melo*	*melo*	inodorus	18	15
			cantalupensis	3	1
			flexuosus	3	1
Cucurbita	*moschata*			3	3

**Table 2 plants-12-03773-t002:** Evaluation by conventional PCR of ToLCNDV presence in leaves, flowers and seed tissues of cucurbit plants. Cantalupensis group (Cant.), flexuosus group (Flex.), Internal control not detected (nd).

		*C. melo* subsp. *melo*																	*C. moschata*		
		Inodorus Group															Cant.	Flex.			
Sample		1	2	3	4	5	6	7	8	9	10	11	12	13	14	15	1	1	1	2	3
Leaves		+	+	+	+	+	+	+	+	+	+	+	+	+	+	+	+	+	+	+	+
Flowers	Petals ♀	+	+	+	+	+	+	+	+	+	nd	+	+	+	+	+	+	+	+	nd	+
	Ovary–pistils	+	+	+	+	+	+	+	+	+	nd	+	+	+	+	+	+	+	−	−	+
	Petals ♂	+	+	+	−	+	+	+	+	+	nd	+	nd	+	+	nd	+	+	+	−	+
	Anthers	+	+	+	+	+	+	+	+	+	nd	+	nd	+	+	+	+	−	+	+	+
Seeds	Coat	+	+	+	+	+	+	+	+	+	+	+	+	+	+	nd	nd	nd	+	+	+
	Endosperm and embryo	+	−	−	+	−	+	−	+	+	+	+	−	+	+	+	−	+	−	−	−

^+^ ToLCNDV detected; ^-^ ToLCNDV not detected; ^♀^ female petals; ^♂^ male petals.

**Table 3 plants-12-03773-t003:** Evaluation of ToLCNDV presence in commercial seeds of watermelon (*C. lanatus*), melon (*C. melo*), cucumber (*C. sativus*) and zucchini (*C. pepo*): Type of cultivar, number of analyzed varieties, number of positive varieties where ToLCNDV was identified in seed and proportion of positive out of analyzed seeds.

Species	Type	Varieties Analyzed	Varieties Positive for ToLCNDV	Positive Seeds/ Analyzed Seeds	Offspring Seedlings Positive for ToLCNDV
Watermelon (*Citrullus lanatus*)	Mini	1	-		
	Black	10	-		
	Striped	19	-		
Cucumber (*Cucumis sativus*)	Short	10	-		
	Long	33	1	1/5	-
Melon (*Cucumis melo*)	Blanco	1	-		
	Amarillo	1	-		
	Cantalupo	2	-		
	Galia	8	-		
	Piel de Sapo	5	1	5/5	1
Zucchini (*Cucurbita pepo*)	Yellow	1	-		
	White	1	-		
	Round	1	1	1/5	-
	Dark green	7	-		
	Edium green	9	-		

^-^ ToLCNDV not detected.

## Data Availability

All data, tables and figures in this manuscript are original.

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
