# Peer review of "Further Molecular Diagnosis Determines Lack of Evidence for Real Seed Transmission of Tomato Leaf Curl New Delhi Virus in Cucurbits"

_plants, 2023, doi:10.3390/plants12213773_

Round 1

Reviewer 1 Report

Comments and Suggestions for Authors

I believe more seeds need to be assayed to determine the true nature of seed transmission for any virus. As such, the sample sizes described in this manuscript appear to be unacceptably small. Other researchers have described experiments using these lower sample sizes but these were from viruses (eg. CGMMV) which are known to be transmitted via seed. The purpose of this manuscript is to rule out seed transmission of ToLCNDV-ES and as such, I think the number of samples is on the small side to make such a claim, despite the various approaches utilized.

Other than the concern with sample size, the data presented are fairly clean and believable.

For the qPCR analysis, no primer efficiencies were stated for the various primers used. This could lead to erroneous conclusions of relative abundance if the viral primers are more/less efficient than the host primers. Have the authors calculated the primer efficiency and incorporated those numbers in the relative abundance calculations?

Comments on the Quality of English Language

This manuscript could use some editing by someone proficient in English as quite a few words are missing/misspelled and some sentences are not complete ones.

Author Response

I believe more seeds need to be assayed to determine the true nature of seed transmission for any virus. As such, the sample sizes described in this manuscript appear to be unacceptably small. Other researchers have described experiments using these lower sample sizes but these were from viruses (eg. CGMMV) which are known to be transmitted via seed. The purpose of this manuscript is to rule out seed transmission of ToLCNDV-ES and as such, I think the number of samples is on the small side to make such a claim, despite the various approaches utilized. Other than the concern with sample size, the data presented are fairly clean and believable.

We agree with the reviewer that a larger number of seeds may provide more robust evidence in understanding the potential seed transmission of viruses. However, the information here provided increases the epidemiological knowledge about ToLCNDV-ES disease in cucurbits and contributes very valuable information to the debate on whether or not ToLCNDV-ES is transmissible by seeds. It is necessary to take into account that in 2020, Kil et al. [37] reported vertical-transmission of ToLCNDV-ES through seeds in zucchini plants (Cucurbita pepo), with transmission rates of overcoming 60% in the evaluated offspring seedlings. However, offspring seedlings were asymptomatic and the only confirmation for viral infection was performed by PCR. Later, Fortes, et al. [42] investigated the possible transmission by melon seeds of a ToLCNDV-ES isolate, and analysis of the progeny plants by hybridization, and conventional PCR or qPCR did not show evidence of ToLCNDV infection in any of the 775 plants evaluated. Here, we have investigated whether the viral particle is transmitted to the offspring seedling by searching for replicative forms of ToLCNDV combining Southern blot hybridization and Rolling Cycle Amplification (RCA). Even though the seedborne nature of ToLCNDV was confirmed, our results using further molecular diagnosis complement those obtained by Fortes et al., and they do not support transmission of this virus from contaminated seeds to the progeny.

Although a larger number of samples assayed in this work and the use of  additional virus diagnostic techniques might be useful to affirm that the virus is not transmitted by seeds, considering the transmission rate of 60% reported for ToLCNDV-ES in C. pepo [37], if we calculate how likely we may be missing the detection of ToLCNDV-ES in the offspring tested of Cucumis melo, Cucurbita moschata and Cucumis sativus in our experiments (n=321, n= 55 and n=10, respectively) the probabilities obtained based on a binomial distribution calculation are about 10−129, 10−22 and 10−5, respectively. These probabilities are even lower if we consider the transmission rate higher than 70% reported for an Asiatic isolate of ToLCNDV in cucumber (C. sativus) by Chang et al. [34]. If we consider the lower infection rate of 25% for an Asiatic strain of ToLCNDV in the progeny seedlings of chayote (Sechium edule, a species of cucurbits distant from those we have evaluated in this study) [33] the probabilities are increased to 10−41, 10−6 and 10−3. Similar probabilities are obtained considering other geminiviruses in different progeny plants, with some seed transmission rates arriving to 85%. Indeed, Tabein et al. [40] confirmed no evidence for seed transmission of tomato yellow leaf curl Sardinia virus in tomato, evaluating 180 seedlings of the progeny (probability of missing the detection of TYLCSV in the progeny was 10−94, considering the 70% seed transmission rate to seedlings previously reported for TYLCV in tomato [30]). Therefore, our experimental design allows to exclude seed transmission of ToLCNDV, at least comparing to the rates that have been reported for ToLCNDV or for other geminiviruses.

For the qPCR analysis, no primer efficiencies were stated for the various primers used. This could lead to erroneous conclusions of relative abundance if the viral primers are more/less efficient than the host primers. Have the authors calculated the primer efficiency and incorporated those numbers in the relative abundance calculations?

We appreciate the query concerning the qPCR analysis, rightly pointing out the importance of primer efficiency in qPCR analysis, especially when comparing the relative abundance of different targets. To address this concern, although the use of these primer pairs had been described in previous publications [45, Sáez et al., 2017 (10.1007/s00299-017-2175-3 )]  here we provide the efficiencies for the primers used in qPCR in our study, both for the viral (Efficiency= 102.65%, slope of the standard curve= -3.26) and host targets (melon: Efficiency= 99.58%, slope of the standard curve= -3.33; Cucurbita: Efficiency= 101.35%, slope of the standard curve= -3.29).

In accordance with the suggestion of all reviewers, the manuscript has been reviewed by a native English speaker

Reviewer 2 Report

Comments and Suggestions for Authors

The authors assessed the distribution of Tomato leaf curl New Delhi virus (ToLCNDV) in leaves, flowers, and seeds from infected plants of susceptible accessions of Cucumis melo and tolerant to infection genotypes of Cucurbita moschata using conventional and quantitative PCR and analyzed whether the viral particle is transmitted to the offspring. Moreover, they evaluated ToLCNDV presence in commercial seeds of cucurbits zucchini, melon, cucumber, and watermelon in their progenies. It was found that integral genomic DNA was not detected in plants of the offspring. Although the seed-borne nature of ToLCNDV has been confirmed, the results of this work do not support the assumption that seed-borne transmission to seedlings of subsequent generations is a likely event for this virus. Even so, the information provided here increases the epidemiological knowledge about ToLCNDV-ES disease in cucurbits, which may be considered to establish phytosanitary policies that guarantee virus‐free plant materials. However, there are many problems in the current version of the manuscript. Some comments can be found below.

1. I suggest that the authors pay more attention to the words they use. Some words in this manuscript are misleading and not precise.

2. Line 51: It should be spread, not introduced.

3. There is supposed to be an improvement in the format of tables 1, 2, and 3.

4. Figure 1 is appropriate, but the characters (A-K) are too small. The parts should be modified.

5. Figure 3 seems to have smaller pixels and is not clear. Please fix it.

6. Following the new rules, virus names should be in block with lowercase. Please revise the whole text accordingly.

7. Please check all the references and ensure they conform to the acceptable formats of the Plants.

Comments on the Quality of English Language

Extensive editing of English language required.

Author Response

  1. I suggest that the authors pay more attention to the words they use. Some words in this manuscript are misleading and not precise.

In accordance with the suggestion of all reviewers, the manuscript has been reviewed by a native English speaker.

  1. Line 51: It should be spread, not introduced. Corrected
  2. There is supposed to be an improvement in the format of tables 1, 2, and 3. The format has been improved
  3. Figure 1 is appropriate, but the characters (A-K) are too small. The parts should be modified. Corrected
  4. Figure 3 seems to have smaller pixels and is not clear. Please fix it. Corrected, quality improved
  5. Following the new rules, virus names should be in block with lowercase. Please revise the whole text accordingly. Revised
  6. Please check all the references and ensure they conform to the acceptable formats of the Plants. Revised

Reviewer 3 Report

Comments and Suggestions for Authors

Plants-2663307 provides some valuable information to the researchers and readers. The subject of the manuscript is consistent with the scope of the Journal. I suggested that the manuscript need to be major revised before it is accepted by this journal.

1.Manuscript needs through language editing.

2.The logic and neat of introduction need to be further improved.

3.Line 46: ‘[1]’ should be superscript. Please check the full text.

4.Line 497: ‘2’ should be subscript. Please check the full text.

5.The conclusion part should be added to make the findings and contributions of the paper clearer.

6.The structural order of the papers is difficult to read.

Comments on the Quality of English Language

Manuscript needs through language editing.

Author Response

1.Manuscript needs through language editing. In accordance with the suggestion of all reviewers, the manuscript has been reviewed and edited by a native English speaker.

2.The logic and neat of introduction need to be further improved.

We have revised the introduction to provide a clearer context for the study and to enhance the logical progression of ideas, providing a more cohesive and structured presentation of the background of the study and making it easier for readers to understand the significance of our work.

3.Line 46: ‘[1]’ should be superscript. Please check the full text.

According to the acceptable formats of Plants, references must be in brackets, without sub- or superscript format.

4.Line 497: ‘2’ should be subscript. Please check the full text. Corrected

5.The conclusion part should be added to make the findings and contributions of the paper clearer.

We have added a Conclusion section to the paper. This section summarizes the key findings of our study and highlights their significance and contributions to the field.

6.The structural order of the papers is difficult to read.

We have reorganized the content of the manuscript to enhance its logical flow. Specifically, we have restructured the introduction to provide a clearer context for the study. In addition, Materials and Methods section is presented now following Introduction section, presenting the research procedures and the results in a more sequential and coherent manner.

Round 2

Reviewer 3 Report

Comments and Suggestions for Authors

I have no comments.

Comments on the Quality of English Language

I have no comments.